# Consumer Knowledge and Acceptance of “Algae” as a Protein Alternative: A UK-Based Qualitative Study

**DOI:** 10.3390/foods11121703

**Published:** 2022-06-10

**Authors:** Chloe Mellor, Rochelle Embling, Louise Neilson, Tennessee Randall, Chloe Wakeham, Michelle D. Lee, Laura L. Wilkinson

**Affiliations:** 1School of Psychology, Faculty of Medicine, Health & Life Science, Swansea University, Swansea SA2 8PP, UK; mellor.chloe@hotmail.co.uk (C.M.); r.j.embling@swansea.ac.uk (R.E.); 908464@swansea.ac.uk (T.R.); 790349@swansea.ac.uk (C.W.); m.d.lee@swansea.ac.uk (M.D.L.); 2BIC Innovation Ltd., One Court Road, Bridgend CF31 1BE, UK; louise.neilson@bic-innovation.com

**Keywords:** algae, macroalgae, microalgae, seaweed, meat substitute, plant-based, alternative protein, consumer acceptance, consumer attitudes, qualitative

## Abstract

Overconsumption of meat has been recognised as a key contributing factor to the climate emergency. Algae (including macroalgae and microalgae) are a nutritious and sustainable food source that may be utilised as an alternative to animal-based proteins. However, little is known about the consumer awareness and acceptance of algae as a protein alternative. The aim of this qualitative study was to develop a rich and contextualised understanding of consumer beliefs about the use of algae in novel and innovative food products. A total of 34 participants from the UK assisted with our study. Each participant engaged in one focus group, with six focus groups conducted in total. Existing consumer knowledge of algae was discussed before participants explored the idea of algae-based food products. Reflexive (inductive) thematic analysis was used to analyse these data. Results showed that consumers have limited pre-existing knowledge of algae as a food source; however, participants were open to the idea of trying to consume algae. This anticipated acceptance of algae was influenced by several product attributes, including perceived novelty, edibility, healthiness, sustainability, and affordability. These findings highlight algae as a promising protein alternative to support plant-forward diets in the UK and identify key attributes to consider in future product development and marketing strategies.

## 1. Introduction

It is widely recognised that the global food system needs to change in response to the climate emergency [1]. Systems supporting the consumption of meat alone are responsible for nearly a fifth of greenhouse gases produced today [2]. Halving meat consumption across Europe would result in a 25–40% reduction in these emissions and adopting “plant-forward” diets in the UK and U.S. alone could account for up to 26% of this goal when compared to the average UK diet [3,4]. However, current trends show that meat consumption continues to increase over time [5]. In the UK in particular, recent figures suggest that reductions in meat intake across consumers remain relatively low [6,7,8], despite 65% of consumers being open to the possibility of moving towards a more sustainable diet [8].

A strategy that has been highlighted to help reduce the consumption of unsustainable meat is the development of innovative plant-based food products and other alternatives to animal-based proteins [9]. The most common alternatives to animal-based proteins currently available on supermarket shelves are beans and lentils (e.g., soya), grains and wheat-based proteins (e.g., seitan), and single-cell proteins (e.g., mycoprotein) [10,11,12]. Despite rapid expansions within this market on a global scale [11,13], acceptance of alternatives to animal-based proteins is often low among consumers, particularly when compared to meat and dairy products [12]. Indeed, it is notable that many consumers remain sceptical about the consumption of alternatives to animal-based proteins. Consumers who do not adopt a plant-based diet often associate reducing their meat intake with poor product appeal, difficulties pleasing the family, decreased protein content, increased costs, and increased effort when preparing meals [14,15]. There are also growing concerns around the potential health impact of consuming such alternatives to animal-based proteins, including the increased salt content of some products [10,16,17], as well as the need to fortify alternatives with essential nutrients to reduce risks of nutritional deficiencies [18].

One alternative to animal-based proteins that has received less attention within this market is “algae”. Macroalgae (commonly referred to as “seaweeds”) and microalgae (such as *Chlorella vulgaris* and *Spirulina platensis*) have been identified as a highly sustainable, nutritious, and versatile food source. This is because algae grow in abundance across coastal regions, which are well adapted to different climates, and do not require agricultural land, fertilisers, or freshwater for farming [19,20]. Species of macroalgae and microalgae are potentially effective sources of protein, with levels reaching up to 47% and 70% of their dry matter [21,22]. Algae are also known to have a number of dietary benefits, such as being low in fat and sodium, and otherwise high in dietary fibres, vitamins, and minerals [22,23,24]. For this reason, algae are often used to fortify products in food manufacturing, in addition to being used as ingredients in a wide array of dishes, such as sushi, soups, salads, breads, snacks, and smoothies [19,25].

Previous research indicates that acceptance of algae as a food source is potentially favourable among consumers in Western countries. Birch, Skallerud, and Paul [26] conducted an online study amongst 502 Australian consumers, finding that 74% had eaten seaweeds in the past, and 62% of consumers were likely to consume seaweeds in the next 12 months. Such acceptability for algae may be driven by the consumer perception and awareness of specific product attributes. For example, quantitative studies have suggested that acceptance may be higher among consumers who report positive beliefs about the taste, freshness, health qualities, and environmental benefits of seaweeds [26,27,28,29]. Individuals also appear to favour purchasing products that are wild-harvested (versus cultivated), have a certification status (e.g., organic), and are locally produced [29,30].

However, food neophobia—the tendency for consumers to avoid consumption of more “novel” foods—has been identified as a significant barrier to consumer acceptance of algae [26,28,31,32,33,34]. One concern is that consumers may be less familiar with seaweed-based food products (with the exception of sushi) [34], and consumers may be unaware of edible uses for microalgae [32]. In the current literature, little is known about the consumer knowledge of algae as a food source. Notably, few qualitative studies have been published on consumer acceptance of alternatives overall [9], with algae being no exception.

Therefore, this study aimed to develop a rich and contextualised understanding of consumer beliefs about algae as a “novel” alternative to animal-based proteins. In a series of online focus groups with UK consumers, we explored the following questions: (1) what the current consumer awareness of algae and algae-based food products is, (2) whether consumers find algae-based food products appealing, (3) whether consumers are willing to try and willing to purchase algae-based food products, (4) how algae compares to other products available in the market from a consumer perspective, and (5) how algae-based food products should be designed and marketed from a consumer perspective.

## 2. Materials and Methods

### 2.1. Participants

The participants (*N* = 34) were UK consumers, with an average of 5.7 participants per focus group. Participants were recruited from Swansea University and surrounding areas, as well as via online platforms (e.g., social media). Eligibility criteria for the study stated that participants must be aged 18+ years, be currently living in the UK, have normal or corrected to normal vision, have no current or pre-existing diagnosis of eating disorders, and have no known food allergies or intolerances. Following recent recommendations for sample size in qualitative studies, six focus groups were conducted to increase the likelihood of reaching data saturation [35], with the aim of including 6–7 participants per group (in line with previous study procedures from our laboratory [36,37]). Participants were overrecruited for the study to account for participant attrition after online sign-up (*n* = 7), as well as participant exclusions (i.e., for failing to use video/microphone in focus groups, *n* = 3). All participants who completed the study received a GBP 10 voucher as compensation for their time. Ethical approval was granted from the Swansea University School of Psychology Ethics Committee (Project ID: 5000).

### 2.2. Focus Groups

Focus group discussions took place between January and February 2021 and were led by the same group facilitator for consistency. All focus groups were conducted using the online video conferencing software “Zoom” (https://zoom.us/, accessed on 3 February 2021), with each focus group scheduled to last up to 60 min. Within these discussions, participants were asked a series of questions about “algae” and “algae-based food products” following a semi-structured interview guide (see Table 1). First, participants were asked to discuss their initial ideas about the term “algae”, with minimal prompts from the group facilitator. Second, participants were shown a short PowerPoint presentation that included additional information about algae as a food source and a range of potential algae-based food products. Participants were then prompted to discuss their willingness to try and willingness to buy algae-based food products, with consideration for product appeal, comparability to other alternative proteins, and potential marketing strategies (e.g., relating to descriptions on product packaging).

### 2.3. Procedure

Participants were directed to sign-up for the study via a brief “Qualtrics” (Qualtrics, Provo, UT, USA) survey, which requested preferred contact information. Participants were informed that the purpose of the study was to investigate “consumer beliefs about a potential new food product”. There was no mention of this product being related to algae or any other alternative to animal-based proteins at sign up. All participants who showed an interest in this study were emailed the consent form (delivered via Qualtrics), which, once completed, allowed them to provide demographic information relating to their age, gender, country of residence, education, and employment. Participants were also asked to provide information about their current diet (“Which option below best describes your current diet?”; tick box options provided for meat, fish, and other animal products and “other” with an accompanying open-text field), the length of time that they had followed their current diet, and reasons for following their current diet in an optional open-text field. After completing their scheduled focus group, participants were emailed a debrief form.

### 2.4. Data Analysis

Focus groups were transcribed from meeting recordings in Zoom, and all participant data were anonymised from the outset. Inductive thematic analysis was conducted based on the guidance proposed by Braun and Clarke [38]. This method of qualitative analysis was chosen because algae as a protein source is a relatively underexplored topic within this literature, and inductive thematic analysis has been described as a sufficient method to use when exploring information about “novel and evolving” research areas [39]. In addition, Braun and Clarke [40] describe “flexibility” as being a key benefit of this approach, allowing for multiple conceptual frameworks to be considered through the analytic process. Initial and secondary coding was carried out using the qualitative data software “Quirkos” (https://www.quirkos.com/, accessed on 11 February 2021). These codes were further developed by being classified into themes and subthemes. A secondary coder was also included in the data analysis process to ensure that themes and codes were accurate and reliable through researcher triangulation. Following a similar approach to that by Puddephatt et al. [41], a random selection of codes (10%) was sent to the secondary coder alongside the developed codebook. The secondary coder was then asked to match codes to what appeared to be the appropriate theme/subtheme. The agreeance rate was satisfactory at 90.23%, and themes were adjusted where applicable in line with subsequent discussions.

### 2.5. Reflexivity Statement

It should be noted that reflexive (inductive) thematic analysis was used in this study [42]. This means that the authors were actively involved in the research process, and that results are not entirely free from potential bias associated with researcher input [38,43]. In line with previous studies using a similar approach [44,45], it is then acknowledged that as authors of this work, we have a general interest in assisting the development of successful and sustainable dietary alternatives to animal-based proteins that may contribute to our interpretation of themes and focus group discussions.

## 3. Results

### 3.1. Participant Characteristics

Participants were aged between 19 and 66 years old (*M* = 34.06; *SD* = 83.27), and included 22 females and 12 males. Participants reported currently living in Wales (*n* = 27), England (*n* = 6), and Scotland (*n* = 1). Participants had completed qualifications equivalent to at least high school (*n* = 2), college level (*n* = 8), HND level (*n* = 3), degree level (*n* = 12), or post-graduate degree level (*n* = 9) and were currently employed (*n* = 18: in a variety of professions including teachers, tradesmen, business owners, assistants, working in admin, etc.), students (*n* = 14), or retired (*n* = 2). Most participants reported consuming foods from a range of food groups, including meats, fish and seafood, and other animal-based proteins (*n* = 21). All other participants reported being pescatarian (*n* = 1; diet includes fish and seafood, but excludes all other meats), vegetarian (*n* = 4; diet excludes meat/fish of any kind), flexitarian (*n* = 6; diet occasionally includes meat/fish) or other (*n* = 2; wholefoods and high protein). Almost all participants reported following their current diet for >10 years (*n* = 28).

### 3.2. Overview of Key Themes

Three key themes were identified within focus groups when discussing algae: pre-existing thoughts about “algae”, product attributes that influence acceptance, and interest in potential food products (see Table 2). Themes and subthemes are described below, with reference to participant quotes from focus groups (anonymised participant IDs are included in brackets).

#### 3.2.1. Theme 1: Pre-Existing Thoughts about “Algae”

Before participants were provided with information about algae, the first identifiable theme of “pre-existing thoughts” highlighted the initial ideas that participants had about the term “algae”. It was evident that participants strongly associated algae with being an aquatic plant, shown through the repeated mention of phrases including “something that grows in water” (FG268), “plant that grows in the sea” (FG224), and “like seaweed” (FG346). Consistent with this viewpoint, many participants imagined algae to be “slimy” (FG379), “green” (FG235), “smelly” (FG109), and “salty” (FG279). Though some participants were generally aware of edible forms of algae (e.g., “there’s one called Wakame” (FG110), “you can get Spirulina powder” (FG154), and “I’ve tried Kelp” (FG143)), few comments related to algae as a potential food source across focus groups within this context.

#### 3.2.2. Theme 2: Product Attributes that Influence Acceptance

Participants often framed algae as a “novel” and “unfamiliar” food source. In the context of discussing edible uses for algae, participants described being willing to try and consume algae if products were appetising, healthy, sustainable, and low cost. However, participants were often unsure about the extent to which algae and algae-based food products would meet these requirements, particularly as the level of processing involved in developing food products increased. Across focus groups, participants wanted to include information about the healthiness and sustainability of products as central features on food packaging, and participants often preferred algae when this was “disguised” as an ingredient within food products.

##### Subtheme: Novelty

When responding to information about algae as a potential food source, willingness to try was most often related to the novelty of consumption. Most participants initially viewed this as a positive feature of the food source, believing that it would create a nuanced experience for them as a consumer and encourage curiosity. However, for those who were not in favour of trying potential food products, comments often related to a lack of familiarity (e.g., “If there wasn’t enough research done”, and “I don’t trust myself to [cook] with that”).

Discussion of marketing strategies for potential algae-based food products also seemed to highlight novelty as a key barrier to consumer acceptance. Participants discussed algae as a food source that “people aren’t really familiar with” and stated that consumers need to “even know that [algae] is an option” by increasing availability. Participants also stated that having limited knowledge on how or what to cook with algae would be a concern when purchasing food products. For this reason, the idea of including recipe ideas and instructions for use on product packaging was proposed as a way to encourage purchasing behaviours.


*“I think people would try it out of curiosity as well, because I would, I’d try it” (FG279)*



*“I would definitely be willing to try it… Because there’s a fun element, isn’t it, there’s a novel element in trying something new” (FG093)*



*“… Having never heard of it, I wouldn’t really know what to do with it, so I probably wouldn’t buy it. I’d think ‘Ah I don’t really know what to do with that’” (FG379)*



*“*
*And I think as well, you’d have to promote recipes… [because] I wouldn’t know what to do with it, I wouldn’t know how to make it into a meal, so I think… you’d obviously have to promote recipes then to give people ideas on how to use it” (FG279)*


##### Subtheme: Edibility

It was evident that participants placed high importance on the appeal and edibility of algae-based food products and that participants wanted products to look and taste aesthetically pleasing. However, it was apparent that participants had some concerns about the “tastiness” of algae. In particular, participants stated that terms such as “algae” and “seaweed” would increase scepticism about the taste and appeal of algae as a food product, due to connotations associated with pre-existing thoughts of “algae”.

In terms of promoting algae-based food products to consumers, participants most often discussed strategies that would mask the use of algae within food products. For instance, participants suggested using algae as an ingredient within other food products and dishes to disguise taste. Relating to product packaging, participants also tended to focus on drawing comparisons to meat and dairy products (e.g., to use terms such as “burger” in product names, and to mimic the appearance of animal-based protein products).


*“…It needs to kind of look appetising or be presented in a way that you know looks nice, smells nice …that’s something you’d want to eat” (FG048)*



*“Is the reason why [algae] hasn’t been [used] because it doesn’t taste very good” (FG313)*



*“I think the thought of the fact that it is algae is probably a lot worse than if somebody hadn’t told you what it was and you’ve tried it” (FG132)*



*“…or whether you disguised it as something [for] like the meat or dairy consumers, [use those] type of like colours and that type of marketing” (FG562)*


##### Subtheme: Healthiness

Participants expected algae to be a healthy food source, shown through the mention of factors including “protein”, “amino acids”, “omega threes”, “high in iron”, “natural”, and “superfood”. Some participants specifically highlighted the use of algae as an alternative for vegetarian and vegan consumers. As such, it was clear throughout focus groups that participants would be more likely to try algae-based food products if the health aspects of the foods were made evident, such as by stating that “health benefits should be used” on the advertisement of products, in addition to highlighting “nutritional values”. However, participants expected the healthiness of algae to decrease if products were processed, particularly when used as an additional ingredient in food products.


*“…I am vegetarian so I’m kind of aware of you know… that there’s not so many plant-based foods that are rich in Omega threes and so for me personally, that would interest me too” (FG048)*



*“… Cornflakes are a great example. You know, they probably started off very healthily, but they actually remove the most nutritious part to produce cornflakes and add lots of things that aren’t healthy, so you know, I think that it would depend on what happened in between …the natural states, and then you know [how] it was produced” (FG048)*



*“But I don’t know how [many] nutrients it loses by the drying process” (FG110)*


##### Subtheme: Sustainability

Participants generally appeared to be uncertain about the sustainability of algae. On the one hand, participants tended to associate algae itself with being “naturally sourced”, “not processed”, and “abundant”. For specific products, participants wanted to know “where it’s come from, how sustainable it is”, and highlighted “local” as a key term relating to sustainability that would be appealing on product packaging. For example, participants described being more likely to purchase algae from local sellers rather than from supermarkets. Participants often stated that key words and phrases such as “Welsh” and “sustainability”, and “contains locally sourced organic seaweed” would likely encourage purchasing behaviour for algae-based food products.

On the other hand, participants often questioned the potential environmental impact of producing algae for consumption, particularly if consumer demand were to increase. For instance, participants wanted to know whether algae would be factory-farmed or wild-harvested and were concerned that the cultivation of algae could potentially impact natural ocean ecosystems. Participants also wanted reassurance about the sustainability of the wider food supply chain, particularly if algae were to be used within processed food products that may require additional food technologies and ingredients.


*“… It’s abundant as well, because you see it in the sea, washed up on the seashore a lot” (FG330)*



*“… I would be more likely to purchase it from somewhere kind of local to here that was more… for example, you know, in one of the local markets they do sometimes at a weekend or something like that, you know if there was a product that had [algae] in it and it looked nice I would probably be more likely to try it there than [I] would in [supermarket name]” (FG048)*



*“Yeah, is it something that’s gonna be grown in a factory or is it gonna be from its natural resources” (FG280)*



*“… It is such an easy product [to] grow locally and sustainably in some ways. But, that being said, if you put it in a protein bar or in a burger or something… even if the algae is sustainable …is the rest of the process sustainable and local as well, or [does] that kind of defeat the purpose of getting your stuff from other places” (FG481)*


##### Subtheme: Affordability

Similar to the discussion of sustainability, participants were divided on the topic of affordability for algae-based food products. Some participants expected items to be reasonably priced, because they believed algae to be available in “abundance” and that producing algae for consumption would not require excessive “energy”. However, many individuals thought that algae-based food products would be expensive to purchase due to their “nuance”. Participants also believed that price would be governed by the specific product being made (e.g., due to the level of “processing” and additional “ingredients” required), as well as the locations in which algae was likely to be sold (e.g., health food stores).

In terms of acceptability, the general trend amongst most participants was that consumers would be less likely to purchase products if they were more costly, “if it was a really expensive product… I’d be less inclined to try”. This was particularly important for those shopping for a family, as participants were concerned about following a food budget and about large quantities of waste should foods be disliked/uneaten.


*“Well, I think when something’s new it’s probably more expensive, but then as it hits the market, and [gets] all the productivity, it [would] probably go down in price” (FG086)*



*“…If you go to just [the local] market, I’m pretty sure you can buy lava-bread and equivalent sort of products, probably reasonably [cheap], but you know if you did go into a healthy shop or you know a sort of trendy food market… then I imagine you know those products would be a lot more expensive” (FG048)*



*“I guess some [products] have [gone] through a lot of processing, so that’s probably quite expensive” (FG143)*



*“It could be very expensive for a small thing... my family won’t necessarily [like] them either, so it’s probably just going to be me, so I don’t really want to be spending a lot of money on it” (FG...)*


#### 3.2.3. Theme 3: Interest in Potential Food Products

There were few direct comparisons between algae and other alternatives to animal-based proteins available on the market, specifically. Rather, participants seemed to focus more generally on the types of products that they would find most appealing and highlighted several examples that were consistent with uses for other alternatives to animal-based proteins. These products included sweet snacks, such as “algae chocolate”, “pudding”, and “mousse”; seasonings, “like you know how you add parsley”; and main dishes and entrees including “a burger”, and “something like [a] cannelloni where it’s mixed into the sauce”. Furthermore, there was mention of algae being used as a sea vegetable in salads, “some kind of like side”, “as a salad equivalent”; and in drinks such as “smoothies” and sodas, “if they can do that, basically [put] two plants into pop [Dandelion and Burdock], maybe [that’s] something we can do with algae”. This was in addition to use of seaweeds as an ingredient within “sushi”.

Despite differences among these proposed ideas, there was a general theme of participants being more willing to purchase algae if it were included as an additional ingredient within existing and familiar products or as a supplement to other dishes. The only exception to this trend was that some participants appeared to prefer the idea of consuming algae in states that were closer to its raw farm. Within this context, participants described being less likely to purchase a highly processed product. Some participants were also less accepting of algae when framed as an ingredient in specific “mock” meat substitutes.


*“Yeah probably, if it was in something I recognized or something like I know I’d eat, then I’d probably buy it then. I’m not sure if I’d just buy it on its own or… I don’t know if you can even eat [algae] on its own, but yeah” (FG268)*



*“I think there’d be a noticeable difference in… any kind of [raw] forms, as opposed to you know [when algae is] used in processed foods, where it’s not kind of obviously there and the taste is probably going to be [flavoured] by something else” (FG048)*



*“I’d be more worried about whether they’re going to use the algae to… make up fake meats and stuff” (FG143)*


## 4. Discussion

This qualitative study aimed to develop our understanding of consumer beliefs about algae as a “novel” alternative to animal-based proteins in the UK. Three main themes and five subthemes were identified using inductive thematic analysis. Across these themes, participants were found to have a limited pre-existing knowledge of algae as a food source (Theme 1). Though participants were generally open to the idea of trying a range of potential algae-based food products (Theme 3), acceptance of algae was influenced by several product attributes, including perceived novelty, edibility, healthiness, sustainability, and affordability (Theme 2).

Our results provide a novel insight into the current consumer awareness of algae, particularly the impact that sensory expectations of new or unfamiliar foods have on consumer acceptance. “Pre-existing thoughts about algae” were consistent with quantitative studies highlighting “algae” as a novel food source to consumers [32,46], as participants had a poor knowledge of edible uses for algae. Results highlighted that consumers may associate algae-based food products with negative connotations that reduce willingness to consume, particularly as this relates to expectations about “edibility” as a key driver of acceptance [28,34,47]. Previous research has shown that when seaweeds are included in higher concentrations within breads (8%), consumers report perceptions of “saltiness” and “strong aftertaste” as attributes that reduced liking when tasting products [48]. Similarly, pasta dishes that are strongly associated with the expected attributes of spirulina have also been shown to be less liked (e.g., “earthy” odour and dark colour) [33]. Given that food neophobia has been identified as a significant barrier to consumer acceptance for algae across studies [26,28,31,32,33,34], such findings further emphasise the importance of identifying strategies to increase consumer familiarity with algae as an alternative to animal-based proteins, particularly as this relates to sensory expectations about foods (e.g., a “salty” taste or “pungent” smell).

The current findings highlighted the importance of considering the use of algae within food products compared to consumption as a standalone food product. Indeed, one strategy that has been discussed in the literature to increase product appeal is to use algae within other food products that are familiar to the consumer (e.g., a recognisable product type or blended with other well-known additional ingredients) [26,49,50]. Our participants specifically commented on the appeal of “disguising” use of algae, and highlighted marketing strategies that increased similarity to other plant-based and animal-based proteins to increase consumer acceptance. This likely reflects concerns about the edibility of products, as previous research has shown that consumer acceptance of an algae-based dish is higher when additional ingredients are more successful at masking the taste and use of algae during consumption [33]. As such, it is notable that participants were generally open to trying a wide range of food products, including both sweet and savoury dishes, snacks, seasonings, and beverages. This is consistent with the current market availability of algae-based food products [19,25], and previous research has also highlighted a similar range of products that consumers deem to be acceptable [47,49]. However, it should be acknowledged that quantitative studies suggest that some of these products may be less acceptable than others across consumer samples [34,47,49]. It is also important to consider whether such product types compliment the sensory characteristics of algae if it is to be used in amounts that may benefit health, given that stronger intensities are less appealing to consumers [33,48]. As such, a specific focus on product sampling may be a key component in the marketing and development of future products, and identifying innovative methods to facilitate more palatable and convenient uses for algae within foods may be particularly helpful (e.g., use of flakes as a “flavour agent” or flours as a “bulking agent” [51]).

Notably, participants discussed a desire for “transparency” about the healthiness and sustainable qualities of algae. Though participants perceived algae to be a natural and abundant resource, some participants were concerned about the effects of food processing on healthiness and sustainability in particular. Additionally, it was highlighted that including nutritional and environmental benefits of algae on product packaging would increase consumer acceptance. This reflects current discussion in the literature regarding the categorisation of many alternative products as “ultra-processed” foods [17] (though this has been debated [16]) and findings of previous research that have identified a consumer preference for purchasing seaweeds when framed as “local” and “wild-harvested” produce [29,30]. This suggests that providing consumers with more information about healthiness and sustainable practices (e.g., on food labels)—and greater consideration for preserving the healthiness and sustainable qualities of foods—could mitigate concerns about product processing from a consumer perspective.

The affordability of algae and potential algae-based food products was highlighted as a potential barrier to consumer acceptance. Participants appeared hesitant to spend on algae as a “novel” food product, mainly due to expected liking. Previous research has shown that consumers are less willing to pay for seaweeds relative to other seafoods [30], and affordability (in light of expected consumer acceptance) may be a wider consumer concern for alternatives to animal-based proteins [14,15]. This suggests that there may be a potential trade-off for consumers when considering product appeal/novelty and perceived affordability. Future research on the balance between price and appeal using a conjoint analysis approach (from an industry and consumer perspective) may be particularly useful to identify how to balance desired product features appropriately [52].

This study provides rich qualitative data for understanding the consumer acceptability of algae-based foods. However, there are some limitations of this approach to consider. First, participants were provided with some contextual information about potential product attributes during focus groups that related to the healthiness and sustainability features in particular and this information also tended to focus on macroalgae. This could have biased perceptions of algae given that initial consumer knowledge was low. We do note that participant discussions diverged from topics introduced in the interview schedule, as participants highlighted additional barriers and concerns associated with consuming algae. However, future research should consider how the provision of such information may impact consumer acceptance for algae, especially as this relates to food labelling.

Second, despite prompts from the group facilitator as part of the interview schedule, participants did not directly compare algae with other alternatives available on the market (e.g., by contrasting expectations about algae with prior experience of plant-based or single-cell proteins). We recognise that it is increasingly important to contrast different alternative proteins in choice-based paradigms, as this can help identify additional traits associated with consumer acceptance [9]. However, we speculate that this reluctance to discuss algae in relation to other alternative proteins is likely underpinned by a lack of consumer awareness, which may account for the gap in consumer acceptance reported for algae versus other plant-based proteins [53]. It may be useful to consider education about alternative proteins (particularly algae) as a nudging strategy that may increase adoption of plant-forward diets in the UK [50]. For example, consumers may be provided with recipes to address uncertainty about how to cook/prepare foods.

Finally, whilst a representative sample was not sought due to the nature of our methodology, most participants were resident in Wales, specifically. This is one region of the UK that is typically known for its continued use of seaweeds within traditional dishes (e.g., “laverbread”) [19,54]. Though this appears to conflict with findings reported on consumer awareness of algae within this study, one media commentary notes that consumption of algae (particularly seaweeds) within Wales has shifted from being a “working-class” staple to a more expensive “superfood” in recent years [55]. This suggests that consumption of algae in such regions may have become more specialised and further emphasises the need to consider the current cultural context within which foods will be consumed when exploring consumer acceptance in future research.

Nevertheless, the current findings highlight potential strategies to encourage the consumption of algae, particularly with respect to the sensory appeal and novelty of products, consumer concerns about health and sustainability, and product affordability. We note that novel and innovative food products—that aim to utilise specific health and sustainability benefits of algae—are already being explored within the food industry (on both a smaller and larger scale). For example, this includes “seaweed flakes” as an alternative to salt, algae-based natural food dyes as an alternative to synthetic food colourings, seaweed sheets as an alternative edible wrap for foods, and algae-based powders as an alternative to eggs in baking [51]. Such products demonstrate how algae consumption (including from processed sources) could be increased within foods whilst considering specific food choice motivations highlighted in this study, though further research is needed to contrast specific nutritional profiles between products.

## 5. Conclusions

This study provides further insight into the knowledge and acceptance of algae amongst a sample of UK consumers. Findings suggest that algae remain a novel food source. Though consumers appear to be open to consuming algae-based food products, increasing awareness and knowledge of the benefits of consuming algae is needed to mitigate concerns relating to expected taste, food processing, and familiarity. For example, using marketing and advertisement strategies, including the provision of recipes and example usage, communicating the benefit of an increase of healthy and sustainable foods within a consumer’s diet. Furthermore, marketing strategies could incorporate product sampling to reduce the barriers relating to the willingness to purchase. Additionally, despite the variation in expected price, it appears that appropriate pricing needs to be considered, in order to increase the likelihood of healthy and sustainable food consumption.

## Figures and Tables

**Table 1 foods-11-01703-t001:** Interview schedule for semi-structured focus groups.

Discussion Phase	Questions	Additional Information Provided to Participants
Warm-up questions	Can you explain what is meant by the term “algae”?	N/A
	Are you aware of the use of algae as a food product? Any examples?	
	What do you think algae-based foods would taste like?	
	Do you think there are any potential benefits of using algae as a food product? Why?	
Appeal of algae-based food products	How appetising/healthy/sustainable/affordable do you think algae-based food products are? Why?	“Algae” are a type of low-energy aquatic plant that has been found to be high in protein, vitamins, minerals, fibre, and fatty acid. There are many types of algae that can be included in food products. This includes “laver”, “kelp”, “wakame”, “ogo”, “sea grapes”, and “mozuku”. A more common name for algae is “seaweeds”. Algae can be included in foods as an alternative source of protein. Some examples of foods that are algae-based include sushi, smoothies, energy bars, meat-free burgers, pasta, and condiments.
Willingness to try and purchase	How willing would you be to try/purchase algae-based food products? Why?	
	Are there any reasons why you may not want to eat algae-based food products?	
Comparability to other alternative proteins	Are you aware of any other alternative-protein foods that you can purchase?	
	How do you think algae-based products compare to these other alternative-protein foods?	
Opinions on product development and marketing	What information do you think should be included on product packaging?	Example descriptions of food products shown to participants, including different terms for “algae”, as well as potential benefits of consuming algae from a health/environmental perspective (e.g., “Contains omega-3 rich algal-oil”, “Kelp noodles”, “Contains locally sourced organic seaweed”).
	Which product description do you prefer? Why?	
	What types of food/beverage products would you like to try that contain algae?	
Closing	Any other thoughts?	N/A

**Table 2 foods-11-01703-t002:** Themes, subthemes, and interconnections between topics discussed in focus groups.

Themes	Sub-Themes	Topics Discussed	Interconnections
Pre-existing thoughts about “algae”		Associations with aquatic plants, edible forms	Novelty, edibility
Product attributes that influence acceptance	Novelty	Familiarity, availability, recipe ideas	Pre-existing thoughts about “algae”, interest in potential food products
	Edibility	Taste, associations with aquatic plants, use within food products	Pre-existing thoughts about “algae”, interest in potential food products
	Healthiness	Nutrition, food processing, use within food products	Interest in potential food products
	Sustainability	Locally sourced, food processing, associations with aquatic plants	Pre-existing thoughts about “algae”, interest in potential food products
	Affordability	Availability, food processing, associations with aquatic plants	Pre-existing thoughts about “algae”, novelty, interest in potential food products
Interest in potential food products		Use within food products, food processing	Novelty, edibility, healthiness, sustainability

## Data Availability

The data presented in this study are available on request from the corresponding author. Full transcripts are not publicly available in order to ensure participant privacy.

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
