# Peer review of "Consumer Knowledge and Acceptance of “Algae” as a Protein Alternative: A UK-Based Qualitative Study"

_foods, 2022, doi:10.3390/foods11121703_

Round 1
Reviewer 1 Report
The objective of this paper is to develop a rich and contextualised understanding of consumer beliefs about the use of algae in novel and innovative food products. Results showed that consumers have a limited pre-existing knowledge of algae as a food source, but that participants were open to the idea of trying to consume algae. This expected acceptance of algae was influenced by several product attributes, including perceived novelty, edibility, healthiness, sustainability, and affordability.
Major Comments
The paper attempts to discuss an important issue. That being said, the paper unfortunately falls short in some ways. In particular, there could be some problems with the designed survey itself, which in turn render relevant discussions and conclusions less credible.
1. An important reason for doubting the conclusions of this paper is the sample size and participant characteristics. Compared to the sample size of some references in this paper (e.g., a total of 3084 Spanish consumers in Lafarga et al. (2021); a total sample of 940 consumers in Weinrich and Elshiewy (2019)), only 34 samples are included in this paper (although it is a qualitative study). Another ambiguity is the description of participant characteristics. The specific occupation and professional knowledge of the participants are not introduced, which could be an important factor affecting consumers' acceptance of algae food from the reference.
2. There is no doubt that the discussion on key themes helps to understand consumers' perception and acceptance of algae food, but its practical significance and value are very confusing. This paper lists some factors that affect the acceptance of algae food, but it is impossible to know how much and why. At least such a discussion is inferior to the quantitative analysis in some references.
3. In the part of conclusion, the authors emphasize that this paper provides further insight into the knowledge and acceptance of algae amongst UK consumers. However, no matter from the breadth or depth of this study, it is impossible to reach the conclusions which are adequately representative of UK consumers in this paper.
Author Response
Reviewer comment: There is no doubt the discussion on key themes helps to understand consumers’ perception and acceptance of algae food, but its practical significance and value are very confusing. This paper lists some factors that affect the acceptance of algae food, but it is impossible to know how much and why. At least such a discussion is inferior to the quantitative analysis in some references.
Author response to reviewer introductory comment and comment 2: We thank the reviewer for their comments. We suspect that we as researchers are coming from a different epistemological position to the reviewer. We feel that it is important to note that as a research group we value both qualitative and quantitative methodologies and view them as complimentary to one another. Indeed, this study represents the qualitative element of a broader project which also led to the publication of a quantitative study which can be found in the journal Food Quality and Preference (that we refer to in the manuscript).
Reviewer comment: Compared to the sample size of some references in this paper (e.g., a total of 3084 Spanish consumers in Lafarga et al. (2021); a total sample of 940 consumers in Weinrich and Elshiewy (2019)), only 34 samples are included in this paper (although it is a qualitative study). Another ambiguity is the description of participant characteristics. The specific occupation and professional knowledge of the participants are not introduced, which could be an important factor affecting consumers’ acceptance of algae food from the reference.
Author response to reviewer comment 1: We understand the reviewers’ first comment regarding sample size and note that as mixed-methods researchers we are mindful around conversations relating to sample size and statistical power. However, for this qualitative study, we have observed best practise for sample size decision making for this methodology and discuss this within the manuscript. As noted in the methodology section of this paper at lines 93-96 and we followed published guidance (Reference 35), that is specifically relevant to qualitative research and focus groups, and therefore ensured an adequate and appropriate sample size.
Additionally, we are grateful to the reviewer for the suggestion to introduce the occupation of the participants. This information was indeed collected, but not included in the paper for the sake of brevity, however has now been detailed between lines 168 – 170.
Reviewer comment: In the part of the conclusion, the authors emphasize that this paper provides further insight into the knowledge and acceptance of algae amongst UK consumers. However, no matter from the breadth and depth of this study, it is impossible to reach the conclusions which are adequately representative of UK consumers in this paper.
Author response to reviewer comment 3: We are grateful to the reviewer for suggesting a rewording of the start of the conclusion, in order to avoid confusion over how representative we are claiming our results to be. This has now been amended and can be found on line 507. We thank the reviewer for bringing this to our attention but would like to take this opportunity to acknowledge that despite making this clearer in the paper, this is not typically something that is traditional in the qualitative methodology that we employed in this study, something we have now made clearer on lines 482 - 483.
Reviewer 2 Report
The article is about consumer knowledge and acceptance of foods with algae as protein. It is a well written manuscript. I read it with interest. I have made some minor suggestions. I hope you find them helpful:
L18, this sentence is hard to understand - six groups with 34 participants each? Six groups with the same participants? I think 34 is the total number of the six groups. Could you clarify that?
The introduction is very clear.
L125 - I would like to see more information on this section. How were the diet and food consumption responses measured? Were all participants who filled out the form invited?
No one said anything about using seaweed to enhance the "fishy" taste in a vegetarian dish?
I missed some suggestions on how we can increase algae consumption if we consider key food choice motivations, e.g., sensory appeal, price, convenience of purchase, etc.
What do the authors suggest practically for algae consumption? The use of algae in industrialized products? Could not that be harmful to health if it comes with more sugar, salt, and fat? Or perhaps the use of algae in culinary preparations (e.g., soup?). If you could share your thoughts with readers, that would be very helpful.
It is a good work, I congratulate the authors.
Author Response
Reviewer comment: L18, this sentence is hard to understand – six groups with 34 participants each? Six groups with the same participants? I think 34 is the total number of the six groups. Could you clarify that?
Author response: We thank the reviewer for identifying the need to add clarity in the abstract with regards to the details on the number of participants and focus group engagement. This has now been amended, lines 8 - 10.
Reviewer comment: L125 – I would like to see more information on this section. How were the diet and food consumption responses measured? Were all participants who filled out the form invited?
Author response: We are grateful for the reviewers’ comment regarding the measurement of participant’s diet/food consumption and acknowledge that it would be beneficial to detail further. This has now been amended in lines 123 – 130.
Reviewer comment: No one said anything about using seaweed to enhance the “fishy” taste in a vegetarian dish?
Author response: Thank you for the interesting question. Despite participants discussing the associations between algae and sea themes (“salty” and “plant that grows in the sea”), there were a lack of comments made by participants regarding using seaweed to enhance the “fishy” tasty in vegetarian dishes. However, there were only 4 participants who reported being vegetarians, which may be a reason why there were a lack of such comments.
Reviewer comment: I missed some suggestions on how we can increase algae consumption if we consider key food choice motivations, e.g., sensory appeal, price, convenience of purchase etc..
Author response: We are grateful to the reviewer for highlighting the benefit that providing suggestions to how algae consumption could be increased with relation to key food choice motivations would have to our paper. We have now highlighted key examples for increasing consumption in line with these motivations throughout (lines 432 – 436, 447, 480-481), and included further discussion of potential food products/ strategies in lines 492 – 503.
Reviewer comment: What do the authors suggest practically for algae consumption? The use of algae in industrialized products? Could that not be harmful if it comes with more sugar, salt and fat? Or perhaps the use of algae in culinary preparation (e.g., soup). If you could share your thoughts with readers, that would be helpful.
Author response: We would like to thank the reviewer for the suggestion to share our thoughts on the practicality of consuming algae-based food products. We acknowledge that the impact of processing on the benefits of alternative proteins is currently debated (see lines 438 – 446). However, we also note some examples of products already being developed by industry to potentially utilise benefits of algae (particularly relating to health and nutrition). Discussion of these examples can now be found on lines 492-503.
Reviewer 3 Report
Dear Authors,
Congratulations on your research on the field of alternative foods. Reducing meat consumption and replacing it with other alternative sources of protein is important not only because of its negative impact on the climate, but also, and perhaps most importantly, on human health. The use of algae seems to be an interesting consideration for wider use as another alternative to meat-based protein. Algae, in addition to being high in protein, are high in fiber, which is also valuable to human health.
The entire manuscript is correctly written. The research conducted, the analysis of the results, the presentation of the results, and the content of the manuscript are unobjectionable.
Below are my suggestions / comments:
Line 166-171 Number of participants by education does not add up to 34 (least high school (N = 10) or degree level (N = 23)). Similar situation is “Most participants reported consuming foods from a range of food groups, including meats, fish and seafood, and other animal based proteins (N = 25). All other participants reported being pescatarian (N = 1; diet in cludes fish and seafood, but excludes all other meats), vegetarian (N = 4; diet excludes meat/fish of any kind), or flexitarian (N = 6; diet occasionally includes meat/fish)”.
Author Response
Reviewer comment: Line 166 – 171: Number of participants by education does not add up to 34 …
Author response: Thank you so much to the reviewer who brought up this point regarding the demographic information not adding up to the total participant number (34). These details have now been amended (lines 166- 170). Additionally, we would like to thank the reviewer for the positive comments and feedback on this paper.
Reviewer 4 Report
The manuscript reads well and provides meaningful qualitative insight on how algae can be introduced amongst UK consumer. Minor comments
Section 2.1
Ethics approval needs a reference number.
Section 2.5
Is there a justification on why thematic analyses is used rather than other approaches?
Discussion - perhaps subsection the paragraphs based on the key highlights of their results.
Author Response
Reviewer comment: Ethical approval needs a reference number.
Author response: We thank the reviewer for bringing the comment regarding the ethical approval reference number for this study to our attention. This has now been included on line 101.
Reviewer comment: Is there a justification for why thematic analyses is used rather than other approaches?
Author response: We appreciate the reviewer for identifying the need to strengthen the justification for using thematic analysis, rather than other qualitative methods. This has now been incorporated into the paper, lines 139 - 141. The additional reference has also been included in the reference list (40).
Reviewer comment: perhaps sub-section the paragraphs based on the key highlights of their results.
Author response: We are very grateful for the reviewers’ suggestion to consider adding subsections to each paragraph within the discussion section, with relation to the highlights of our key results. In the spirit of this comment, we have now signposted the nature of each paragraph, with relation to the significance that our results have on identifying consumer acceptance of algae as a food product. This can be found in lines 398 – 400, 414 – 415, 437 – 438, and 450 – 451.
Round 2
Reviewer 1 Report
Though the authors largely accomplish what they set out to do, the article can be stronger with a deeper conceptual/theoretical commitment. Moreover, the paper is a straightforward presentation of narrowly conceived and scrutinized data with a literature review that quickly covers the necessary work. However, the literature doesn't seem to inform or permeate the work and simply gives work a thin veneer of justification and the Discussion/Conclusions/Policy proposals are only tangentially related to the analysis. Overall, I insist more analytical rigor needs to be applied to support the author’s quantitative results.